# Mapping IgA Epitope and Cross-Reactivity between Severe Acute Respiratory Syndrome-Associated Coronavirus 2 and DENV

**DOI:** 10.3390/vaccines11121749

**Published:** 2023-11-24

**Authors:** Salvatore G. De-Simone, Paloma Napoleão-Pêgo, Guilherme C. Lechuga, João P. R. S. Carvalho, Maria E. Monteiro, Carlos M. Morel, David W. Provance

**Affiliations:** 1Center for Technological Development in Health (CDTS)/National Institute of Science and Technology for Innovation in Neglected Population Diseases (INCT-IDPN), Oswaldo Cruz Foundation, Rio de Janeiro 21040-900, RJ, Brazil; pegopn@gmail.com (P.N.-P.); gclechuga@gmail.com (G.C.L.); joaopedrorsc@gmail.com (J.P.R.S.C.); monteiro.meduarda@gmail.com (M.E.M.); carlos.morel@fiocruz.br (C.M.M.); bill.provance@fiocruz.br (D.W.P.J.); 2Epidemiology and Molecular Systematics Laboratory (LEMS), Oswaldo Cruz Institute, Oswaldo Cruz Foundation, Rio de Janeiro 21040-900, RJ, Brazil; 3Program of Post-Graduation on Science and Biotechnology, Department of Molecular and Cellular Biology, Biology Institute, Federal Fluminense University, Niterói 22040-036, RJ, Brazil; 4Program of Post-Graduation on Parasitic Biology, Oswaldo Cruz Institute, Oswaldo Cruz Foundation, Rio de Janeiro 21040-900, RJ, Brazil

**Keywords:** COVID-19, SARS-CoV-2, SARS-CoV-2 variants, IgA epitopes, IgA-diagnostic, cross-reactive epitopes, mucosal immunity

## Abstract

Background: The newly introduced COVID-19 vaccines have reduced disease severity and hospitalizations. However, they do not significantly prevent infection or transmission. In the same context, measuring IgM and IgG antibody levels is important, but it does not provide information about the status of the mucosal immune response. This article describes a comprehensive mapping of IgA epitopes of the S protein, its cross-reactivity, and the development of an ELISA-peptide assay. Methods: IgA epitope mapping was conducted using SPOT synthesis and sera from RT-qPCR COVID-19-positive patients. Specific and cross-reacting epitopes were identified, and an evolutionary analysis from the early Wuhan strain to the Omicron variant was performed using bioinformatics tools and a microarray of peptides. The selected epitopes were chemically synthesized and evaluated using ELISA-IgA. Results: A total of 40 IgA epitopes were identified with 23 in S1 and 17 in the S2 subunit. Among these, at least 23 epitopes showed cross-reactivity with DENV and other organisms and 24 showed cross-reactivity with other associated coronaviruses. Three MAP4 polypeptides were validated by ELISA, demonstrating a sensitivity of 90–99.96% and a specificity of 100%. Among the six IgA-RBD epitopes, only the SC/18 epitope of the Omicron variants (BA.2 and BA.2.12.1) presented a single IgA epitope. Conclusions: This research unveiled the IgA epitome of the S protein and identified many epitopes that exhibit cross-reactivity with DENV and other coronaviruses. The S protein of variants from Wuhan to Omicron retains many conserved IgA epitopes except for one epitope (#SCov/18). The cross-reactivity with DENV suggests limitations in using the whole S protein or the S1/S2/RBD segment for IgA serological diagnostic tests for COVID-19. The expression of these identified specific epitopes as diagnostic biomarkers could facilitate monitoring mucosal immunity to COVID-19, potentially leading to more accurate diagnoses and alternative mucosal vaccines.

## 1. Introduction

The humoral response is pivotal in adaptive immunity against numerous viral infections [1]. In COVID-19, alpha and gamma immunoglobulins (Ig) derived from infected individuals or those who have received vaccinations contribute to viral neutralization. However, their roles in immunity differ across various infection stages and specific anatomical sites [2,3,4]. Among these, IgA, predominant in mucous membranes, is the most abundantly produced Ig in humans (66 mg/kg/day), while IgG is the primary isotype in blood and reaches most tissues by diffusion. [2,5]. The distribution of IgA on epithelial mucosal surfaces that encounter infectious agents positions it uniquely for intervening in transmission since complement and phagocytes are not normally present and, therefore, function chiefly as a neutralizing antibodies.

The principal target of human IgG and IgA against the severe acute respiratory syndrome coronavirus 2 (SARS-CoV-2) is the integral spike (S) protein, which is common to all currently employed vaccines [6,7]. The timeline of IgA and IgG responses induced by mRNA vaccines during trials was recently published [8,9]. Notably, IgA against the SARS-CoV-2 spike protein emerges earlier in both infected [7,8,10,11] and vaccinated patients [12,13], demonstrating superior antiviral potency to IgG not only against SARS-CoV-2 but also influenza [4,11,14].

The serum IgA isotype proves to be seven times more effective in viral neutralization than IgG [12], and the IgA dimers, the primary form in the nasopharynx, are approximately 15 times more potent than IgA monomers [15]. Thus, secretory IgA (sIgA) responses may be particularly valuable for protection against SARS-CoV-2 and for vaccine efficacy. This antiviral protective immunity is also evident in the temporal dynamics of circulating IgA+ plasmablasts equipped with mucosal homing receptors and in the presence of neutralizing IgA in airway fluid and saliva [10,11]. Conversely, the overall levels of immunoglobulins (IgA, IgG, and IgM) and complement proteins (C3 and C4) in COVID-19 patients have been found within the normal range [11]. However, significant differences in their persistence in the serum after infection [12,16,17] and COVID-19 vaccination [12,18] have been demonstrated. Elevated IgG or IgM antibody levels targeting SARS-CoV-2 spike protein or receptor-binding domain (RBD) appear ten days after symptom onset. The average antibody response pattern is an early IgM increase followed by IgG development. Although different seroconversion types exist, such as the synchronous seroconversion of IgG-IgM, earlier IgM seroconversion, and delayed IgM seroconversion [19,20,21,22], the clinical value of antibody testing has yet to be fully demonstrated.

The role played by serical and mucosal IgA responses, the total IgA generation rate from this response, and the involvement of its epitopes in COVID-19 severity and/or vaccination are largely poor explored areas [3,4,23] aside from notable instances such as frequent thromboembolisms in severe COVID-19 cases [24,25,26,27].

Davis et al. [28] characterized the IgA immune response regarding neutralization and Fc-effector function. They found that the plasma IgA response contributed to the neutralization antibody response of wild-type SARS-CoV-2 RBA and various RBD mutations despite displaying greater heterogeneity, and it was less potent than IgG.

Several other investigators have also examined various aspects of the IgA immune response to the S protein in the context of COVID-19. They have utilized various techniques, such as studying fragments of peptides with different sizes [29], exploring antibody affinity maturation in relation to clinical outcomes in hospitalized COVID-19 patients [30], employing microarray analysis of peptides to investigate the disease severity over time in a small cohort of patients [23], and using a microarray of peptides technology to analyze the humoral response profile to COVID-19 [31]. 

In this context, some lateral flow rapid tests, enzyme-linked immunosorbent assay (ELISA) kits, and IgA-based assays have been regarded as more sensitive yet less specific than IgG-based counterparts [32,33]. However, cross-reactivity has been detected with both tests [34,35,36,37,38]. However, the continuous emergence of new kits in the market and the potential to develop in-house kits based on the quality of SARS-CoV-2 antigen plates presents various options.

It can be challenging to differentiate viral infections caused by COVID from arbovirus infections, like DENV and CHYV [34,39,40,41], due to their similar clinical and laboratory characteristics. This similarity can lead to the consideration of COVID-19 when, in fact, a false positive dengue serology result may result in a misdiagnosis with potentially serious concerns.

Consequently, this study aimed to map the IgA linear B cell epitopes of the COVID-19 spike protein, encompassing both native and mutated variants. Furthermore, we identify the cross-reactive, pinpoint specific epitopes, and assess these peptides using ELISA, employing sera from COVID-19 and dengue virus-infected patients before the pandemic.

## 2. Materials and Methods

### 2.1. Patient Samples and Ethical Approval

A panel comprising 33 serum samples (Appendix A) was obtained from symptomatic patients with confirmed positive PCR diagnostic tests at the pandemic’s beginning and 25 sera from vaccinated patients. Twenty-four serum samples of patients diagnosed with dengue fever were generously supplied by the Laboratory of Flavivirus of the Oswaldo Cruz Institute of FIOCRUZ in Rio de Janeiro and the Laboratory Central of Public Health of Ceará state, Brazil. In addition, serum from healthy donors, collected before the pandemic, was provided by HEMORIO, which is a centralized network of blood donor facilities in Rio de Janeiro, Brazil. To ensure anonymity, all samples were delivered without any identifying information of the patients.

### 2.2. Synthesis of the Cellulose-Membrane-Bound Peptide Array

The complete sequence (1273 amino acids) of the S protein from SARS-CoV (P0DTC2) available in the UniProt database (http://www.uniprot.org/; accessed on 27 January 2020) was encompassed in the synthesis of 14-residue-long peptides. These peptides, with a 9-residue overlap, were semiautomatically generated on cellulose membranes (Amino-PEG 500-UC540) and using an Auto-Spot Robot ASP-222 from Intavis Bioanalytical Instruments AG, Köln, Germany. The peptide library contained 258 sequences for P0DTC2. Coupling, blocking, and deprotection were performed until the desired peptide was generated and carried out as described previously [42]. Membranes containing the synthetic peptides were probed immediately. Peptides as negative controls were included in each assay.

### 2.3. Screening of SPOT Membranes and Measurement of SPOT Signal Intensities

After the peptide synthesis, the cellulose membranes underwent extensive washing with TTBS (50 mM Tris, 136 mM NaCl, containing 1.5% casein) and were then exposed to a pool of sera from 10 symptomatic patients who tested positive for COVID-19 through PCR (Appendix A). The sera were diluted 1:100 in TBS-CT and incubated with goat anti-human IgA-AP (alkaline phosphatase labeled, diluted 1:5000; Kirkegaard & Perry Lab, Inc., Gaithersburgm, MD, USA) at room temperature for 60 min. Next, the membranes were washed with TTBS containing 0.5 M NaCl, which was followed by CBS (50 mM citrate-buffer saline). Finally, the Chemiluminescent CDP-Star^®^ Substrate (0.25 mM) with Nitro-Block-II™ Enhance (Applied Biosystems, Waltham, MA, USA) was added to initiate the reaction. The chemiluminescent signals were detected on an Odyssey FC (LI-COR Bioscience, Lincoln, NE, USA) using previously established conditions [43]. To quantify the signal intensities, a digital image file was generated at a resolution of 5 MP, and the signal intensities were quantified for 2 min using Total Lab TL100 (v2009, Nonlinear Dynamics, Newcastle, Tyne, UK) software. An epitope was defined as the minor resultant sequence of two or more positive contiguous spots with a signal intensity (SI) greater than or equal to 30% of the highest value obtained from the set of spots on the respective membrane. The signal intensity (SI) used as a background was derived from a set of negative controls spotted on each membrane. Finally, a comparative analysis of the reactivity index of the spots, normalized on a dimensional hierarchical level, was conducted using the approach previously described [44].

### 2.4. Structural Localization of the Epitopes

The tridimensional structures of spike protein were obtained from I-Tasser (https://zhanggroup.org/I-TASSER/, accessed on 22 July 2022) and subsequently visualized and analyzed utilizing the PyMOL Molecular Graphics System, Version 2.0 by Schrödinger, LLC. The models were selected based on the optimal C-score and TM-score (topological evaluation value) [45]. The secondary structure of the proteins was predicted by the [http://bioinf.cs.ucl.ac.uk/psipred/, accessed on 22 July 2022] and CDM [https://cybertaxonomy.org/cdmserver/installation, accessed on 22 July 2022]. 

For comparative analysis, the receptor-binding domain motif (RBD) sequences from all reported SARS-CoV-2 variants of concern were retrieved from the viralzone database (https://viralzone.expasy.org/, accessed on 22 July 2022). The sequences of identified IgA epitopes were aligned using CustalW on MEGA11 software. IgA epitope sequences from all variants of concern were further analyzed by sequence logo using Weblogo (https://weblogo.threeplusone.com/, accessed on 22 July 2022) to evaluate physicochemical and biochemical changes in mutated sequences.

### 2.5. Preparation of the MAP Peptides

The dendrimer multi-antigen peptides (MAP4s) were used for preparation using the tetrameric Fmoc2-Lys-B-Ala Wang resin (CEM, Corp, Charlotte, NC, USA). The peptides possessed the sequence GSYADSFVIRDGSGS (pep-248; epitope #SC/14; aa395–404), GSNNSNN DSKVGSGS (pep-249; epitope #SC/16; aa 436–449), and GSLKPFERDISTGSG (pep-250; epitope #SC/17; aa459–468). In each sequence of the peptides, the dipeptide GS in the N-terminus, the tetrapeptide GSGS (#SC/14 and SC/#15), and the tripeptide GSG (#SC/17) in the C-terminus were used to complete fifteen amino acids. The synthesis was conducted on an automated peptide synthesizer (MultiPep-1 CEM Corp, Charlotte, NC, USA) with F-moc-amino acids and protecting groups as required. The peptides were then cleaved, deprotected, and precipitated, and each MAP4 was analyzed as described previously [46]. In most cases, the peptides were used as prepared.

### 2.6. In House ELISA

The peptide ELISA (pELISA) was conducted as described previously [46]. Immunolon 2HB plates (Immunochemistry Technologies, Bloomington, MN, USA) were coated overnight at 4 °C with 500 ng of peptides per well in coating buffer (50 mM Na_2_CO_3_–NaHCO_3_, pH 9.6). After each incubation step, the plates were washed three times using PBS-T washing buffer (PBS with 0.1% Tween 20 adjusted to pH 7.2), blocked (200 µL) with 1% BSA and incubated for 1 h at 37 °C. Next, the patient’s sera were diluted (1:25) in coating buffer, and 100 µL was applied onto immunosorbent plates and incubated for 1 h at 37 °C. Following several washes with PBS-T, the plates were incubated with 100 µL goat anti-human IgA-HRP (1:5000 dilutions at blocking buffer; Sigma-Aldrich, St Louis, MO, USA) for 1 h at 37 °C. Finally, 3,3′,5,5′-tetramethylbenzidine (1-Step™ Ultra TMB-ELISA, Scienco Biotech Ltd., Lages, SC, Brazil) was added for 15 min, and the reaction was stopped by adding 0.5 M sulfuric acid. The absorbance values at 405 nm were read using an ELX800 Microplate Reader (Bio-Tek Instruments Inc, Winooski, VT, USA). The plate was read within 2 h of adding the stop solution. Values of blank wells, which contained only peptides, were subtracted from the sample’s optic density.

### 2.7. Statistical Analysis

Data were analyzed using GraphPad Prism software (GraphPad version 6, San Diego, CA, USA). The reactivity index (RI) reflected the absorbance divided by the cutoff (mean of negative samples + 2 times the standard deviation) determined by each peptide. All results >1 was considered positive and <0.99 negative. Kruskal–Wallis’s test was applied to identify statistical differences, which was followed by Dunn’s multiple comparisons tests. Significant differences were considered with *p* < 0.05.

## 3. Results

### 3.1. Identification of IgA Epitopes in the COVID-19 Spike Protein

The strategy used to characterize the epitopes is shown in the flowchart presented in Figure 1. IgA epitopes within the COVID-19 Spike protein (1273 aa) were identified through the recognition of peptides in a synthesized library (144 peptides) by sera from COVID-19 patients (Material and Methods). Figure 2A depicts the chemiluminescent signal image obtained from each library peptide, showing their reactivity with human IgA antibodies in pooled sera from infected patients. Figure 2B displays the measured intensity and the position of each peptide. Intensities were normalized to 100%, as established by the positive control. A list of the synthesized peptides and their corresponding positions on the membranes is provided in Appendix A. The antibody reactivity pattern generated in infected patients demonstrated the recognition of many peptides (Figure 2A). An analysis of the sequences comprising the synthesized peptides in reactive regions defined forty major epitopes recognized by patient sera (Table 1).

Figure 2C shows the results of the hierarchic comparison of the reactivity index of the SPOT peptides normalized on a dimensional hierarchical level. The hierarchical levels sort the vertices based on their distance from the initial subgraphs and sort the vertices based on their distance. The graph’s layout shows its overall hierarchical structure with the epitope reactivity index being the epitope with the highest values highlighted in the upper left portion of the image [45]. Thus, the peptide E8, followed by C20 and F6, was the most reactive.

### 3.2. Structural Localization of the IgA Epitopes in the Subunits and RBD of the S Protein

The S protein of SARS-CoV-2 has 1273 aas and plays a key role in the receptor recognition and cell membrane fusion process. It is composed of a signal peptide (aa 1–13) located at the N-terminus, and two subunits, S1 (aa 14–685) and S2 (aa 686–1273). The S1 subunit contains the N-terminal domain (aa14–305) and the receptor-binding domain (RBD, aa319–541) that recognizes and binds to the host receptor ACE-2. In the S2 subunit, there is the fusion peptide (FP, aa788–806), heptapeptide repeat sequence (HR1, aa912–984), HR2 (aa1163–1213), TM domain (aa1213–1237), and cytoplasm domain (aa1237–1273) [47,48].

The crystallographic structure available in PDB (PDB: 1xdt) was used to analyze the location of the epitopes in S protein. It displays the spatial location of the forty IgA reactive epitopes identified by the SPOT synthesis array experiments (Table 1). Most of the identified epitopes were in loop/coil structures, present on the protein surface, and accessible to the solvent. Twenty-three epitopes were placed in the S1 subunit (Table 1, Figure 3), six (SC/13huA to SC/18huA) of which were situated in the RBD of the S protein (Table 1, Figure 3A). Seventeen epitopes (SC/24huA to SC/40huA) were positioned in the S2 subunit (Figure 3C), one was positioned in the fusion peptide (SC/27huA), three were positioned in the HR1 (SC/31huA, SC/32hua, and SC/33huA) and two were positioned in HR2 (SC/39huA and SC/40huA) (Table 1). No epitope was detected in the transmembrane region (aa1213–1237) and the CyD (aa 1237–1273).

BLASTP analysis against non-redundant sequences deposited in diverse databases and containing amino acids aligning without gaps showed twenty-four DENV cross-reactive epitopes (Table 1). These cross-reactive epitopes were distributed in S1 (17) and S2 (15). The S1/RBD contained three cross-reactive epitopes (SC/13, SC/15, and SC/18), the S2/HR1 contained all three (SC/31, SC/32, and SC/33), and the S2/HR2 contained one (SC/40). 

Beta-coronaviruses (beta-CoVs) share striking similarities in genome structure and exhibit immunological relatedness. Consequently, we delved into investigating IgA epitope correlations among different beta-CoVs. Our study unveiled that twelve IgA epitopes within the S1 domain demonstrated cross-reactivity with SARS-CoV. Notably, these epitopes encompassed SC/7, SC/9, SC/11, SC/12, SC/19, SC/20, and SC/23. Furthermore, the cross-reactivity extended to encompass all six epitopes located in the S1/RBD region (SC/13 to SC/18) and the twelve within the S2 domain (SC/25, SC/27, SC/30 to SC/38, and SC/40). Remarkably, this cross-reactivity was observed not only with SARS-CoV but also with MERS-CoV and MERS-OC43. Additionally, epitope SC/32 displayed cross-reactivity with HKU1 (Appendix A).

### 3.3. Evolutionary Analysis of the RBD IgA Epitopes in the Spike Protein of SARS-CoV-2 Variants (from Alpha to Omicron)

The protection of the vaccines against variants of the SARS-CoV-2 is dependent on the common sequences of the epitopes. However, the phylogenetic analysis identified that at least one epitope presented significant structural divergence (Table 2). Therefore, these experiments had two objectives: (1) to validate the six identified IgA epitopes of the RBD and (2) to demonstrate the absence of cross-reactivity of the mutated SC/18huA epitope (Omicron Ba.4 and Omicron Ba.5).

As seen in Table 2, the SC/18huA epitope was the one that showed the greatest divergence, especially in the Omicron variants, which was followed by the SC/16 and SC/14 epitopes. The SC/13 and SC/17 epitopes remained constant in all variants and the wild-type virus (Wuhan). Multiple alignments also showed a significant charge and amino acid composition change in these epitopes (Figure 4).

Thus, to validate and confirm the immunological importance of the mutated epitopes of the new variants, six IgA–epitope peptides (#SC/13huA to #SC/18huA) were synthesized covalently attached on cellulose support (SPOT synthesis) as described before and evaluated using a pool (*n* = 10; Appendix A) of sera from the hospitalized patient who had been independently confirmed with COVID-19 by PCR. Only the shortest epitope sequence was used in the synthesis. However, to maintain a reactive sequence in the assays, all peptides were synthesized with 15 amino acids in length using glycines at the N and C terminals as spacing. Negative controls included collections both before and after the COVID-19 pandemic were included. The list of synthesized peptides is shown in Appendix A.

The immunoreactivity of the patients’ sera collected during the different steps of the pandemic (1st phase, vaccinated, and 2nd phase-mutated COVID) against the epitopes of the non-mutated and mutated epitopes of the RBD is shown in Figure 5. This approach revealed that from the six IgA–RBD epitopes, only the SC/18 of the Omicron variants (Ba.1, BA.2, BA2.12.1, BA.4, and BA.5) presented a single IgA epitope reactive with sera from patients from the second wave (Omicron).

### 3.4. Validation by pELISA of the Three RBD-Specific Epitopes

Before undertaking this analysis, we assessed the potential for cross-reactivity of a commercial IgA-ELISA assay (SARS-CoV-2 IgA; Serion Immunomat, Würzburg, Germany) with our collection of DENV patient sera gathered before the pandemic. The results indicated that 70% of the DENV patient sera (*n* = 24) and 90% of the sera from hospitalized SARS-CoV-2 patients (*n* = 33) tested positive for IgA antibodies (Appendix A). It is important to note that this assay is designed based on the entire S structural membrane protein. 

To validate our chosen RBD epitopes, we compared the same group of DENV patient sera (*n* = 16) and sera from individuals who received COVID-19 vaccinations (*n* = 25). The results of this comparison are displayed in Figure 6.

Significant differences between pre-pandemic DENV-positive and vaccinated samples were evidenced for all peptides. Higher reactivity index and positive samples were evidenced for vaccinated individuals for all three tested peptides: 248 (GSYADSF VIRDGGSG; RI mean = 1.0 ± 0.3, positive samples = 11), 249 (GSNNSNNDSKVGSGS; RI mean = 1.1 ± 0.5; positive samples = 15) and 250 (GSLKPFERDISTGSG; RI mean = 1.5 ± 0.5; positive samples = 21). Pre-pandemic DENV-positive sera did not significantly cross-react for all three tested peptides.

## 4. Discussion

The human immune system can remember various target antigens (epitopes). It can effectively respond to these antigens upon subsequent encounters, provided it has been previously primed. Coronaviruses share enough common features to elicit cross-reactive immune responses despite their genetic diversity. Additionally, the distribution of IgA and IgM on mucosal surfaces exposed to infectious agents uniquely positions the immune system to intervene in transmission. IgA1 is the dominant immunoglobulin in the respiratory tract, and it is the primary entry point for many microorganisms. Consequently, IgA1, including secretory IgA, plays a pivotal role in defending against respiratory pathogens by neutralizing or preventing their attachment to the mucosal epithelium [49].

In this study, we have identified forty IgA epitopes within the Spike protein of SARS-CoV-2, which were recognized by patients’ sera through a peptide microarray analysis. This discovery highlights the Spike protein’s potent mucosal immunostimulatory adjuvant properties, which are capable of eliciting robust mucosal and immune responses. Notably, all existing vaccines primarily target the Spike protein. However, recent research has shown that the SARS-CoV-2-neutralizing IgA response occurs earlier than the IgG response but is modest and diminishes more rapidly [50].

Among these IgA epitopes, eight appeared specific to SARS-CoV-2, while thirty-two exhibited cross-reactivity with proteins from the dengue virus. All forty epitopes were exposed on the molecular surface and accessible to the immune system (Figure 3).

HCoV generally infects target cells through the action of S proteins, which exhibit characteristics resembling class I fusion proteins [47,51]. These S proteins are trimeric integral membrane proteins, structured into subdomains crucial for engaging the receptor angiotensin-converting enzyme 2 (ACE2) (S1/RBD), and harbor an S2/HR1 and HR2 responsible for facilitating membrane fusion [52]. Due to this, the S2 region has been considered an intriguing therapeutic target for addressing COVID-19. Although HR1 and HR2 have been observed to be transiently exposed during the fusion process, notable antibody responses against these S2 regions have yet to be previously reported. Our study successfully identified five IgA epitopes within this region: SC/31, SC/32, and SC/33 in the S2/HR1 segment and epitopes SC/39 and SC/40 in the S2/HR2 segment. All these epitopes exhibited reactivity with DENV except for SC/39, which was specific to COVID-19 (Table 1).

The variability of COVID-19 suggests that an individual’s immune response to SARS-CoV-2 plays a crucial role in determining the clinical course, ranging from asymptomatic to severe disease. In response to pathogens with no pre-existing immunity, our bodies rapidly engage the innate immune response to generate highly specific and effective tools: high-affinity antibodies, B cells, and T cells. Extensive analysis of the antibody response has shown that SARS-CoV-2 induces virus-specific antibodies across all immunoglobulin isotypes, including IgM, IgA, and IgG [19,53].

These isotypes are specialized to function within distinct body compartments. Due to the ability of a particular V region to associate with any C region through isotype switching, a singular B cell’s offspring can generate antibodies specific to the same triggering antigen, which can provide the requisite protective functions for each bodily compartment [54].

In the humoral immune response against COVID-19, the initial antibodies produced are invariably of the IgM class primarily because IgM expression can occur without isotype switching. Despite this, IgM molecules exhibit multipoint binding, increasing overall avidity. Subsequent immune responses also can yield some IgM production after somatic hypermutation, although other isotypes tend to dominate in the later stages of the antibody response [32,55]. Alternatively, the class switching can also result in the formation of IgA [32].

The efficacy of the IgA heavily relies on the affinity of individual antigen-binding sites for their specific antigens. B cells expressing IgA isotypes are typically selected for increased antigen-binding affinity within germinal centers, emphasizing the importance of their precision and effectiveness in immune responses [56]. Thus, the diversity in isotypes and the capacity to modify affinity for epitopes enable the immune system to exhibit greater versatility and efficiency in safeguarding the organism against a broad spectrum of threats. This is also the case of the 2F5 region of gp42 of HIV/AIDS that induces IgA2 and IgG1 antibodies, which act synergistically, blocking the transfer of HIV-1 from Langerhans to autologous CD4+ T cells and inhibiting CD4+ T cell infection [57].

As antibodies are versatile tools in the immune system’s toolkit, each isotype is designed to tackle specific challenges. However, these features can vary, as is the case of different mutations incorporated in the SARS-CoV-2, which can represent a different epitope that creates different antibody isotypes [58]. Each isotype might prefer certain epitopes or residues of amino acids, which may be better suited to deal with specific types of invaders. For example, of the 40 IgA epitopes identified in this study, 25 share amino acids with the IgG isotype. The RBD of the SARS-CoV-2 encompasses six of these IgA epitopes (Table 1; SC/13huA to SC/18huA) and six IgG epitopes (aa355–364, aa395–404, aa415–424, aa440–449, aa460–469, aa490–504) [59]. However, these epitopes possess different residues (right or left) but shared residues indicating distinct preferences of response and evolution of the immune response, providing immunoglobulins with a higher binding or neutralization ability. Therefore, changes in the affinity epitopes will allow the immune system to be highly adaptable and effective in fine tuning, ready to handle diverse threats or situations [60].

Our study also demonstrates that the RBD of all SARS-CoV-2 variants contains six IgA epitopes (three SARS-CoV-2 specific (SC/14, SC/16, and SC/17) and three DENV cross-reactive) with only the Omicron BA.2 and BA.2.12.1 variants possessing a critical epitope capable of inducing a particular IgA response (Figure 4). The other six variants (Alpha, Delta, Gamma, Omicron BA.1, BA.4, and BA.5) exhibit mutation points in the S1/RBD. Still, these alterations do not interfere with IgA binding induced by the ancestral variant (Figure 5 and Appendix A).

Phylogenetic analysis conducted on human beta-CoV reveals a notable observation: many IgA epitopes from SARS-CoV-2 display cross-reactivity with SARS-CoV. This includes all six epitopes within the S1/RBD (SC/13 to SC/18). Furthermore, cross-reactivity was also identified with MERS-CoV and MERS-OC43 (Appendix A).

Regarding cross-reactivity with other organisms, we have identified thirty-two cross-reactive epitopes with twenty-eight interacting with DENV proteins and four interacting with proteins from various organisms (Table 1). Previous studies using rapid IgM and IgG diagnostic assays and ELISAs have shown serological cross-reactivity between DENV, ZIKA, other flaviviruses, and SARS-CoV-2 [38,40,61,62]. Cross-reactivity has also been observed with diagnostic tests for IgA [34,35,36,37,38]. Furthermore, the presence of cross-reactive IgA in the S1 subunit of the spike protein has been detected in the saliva of uninfected individuals living in areas of DENV [63].

As demonstrated in our study, these findings underscore the importance of identifying and selecting specific epitopes to develop more accurate diagnostic tests for SARS-CoV-2. Depending on its purpose, specificity in serological testing is essential whether for prevalence screening or diagnosing individual patients. High sensitivity (≥95%) is critical for diagnosing individual patients, while high specificity (≥98%) is preferred for seroprevalence studies to minimize false-positive results. The choice of test characteristics depends on the pretest probability of the disease.

The results of our study highlight the potential of three selected epitopes (#SC/14, #SC/16, and #SC/17) to effectively distinguish between negative and positive samples in serological diagnosis. These epitopes are suitable for further evaluation in phase IIA studies and support their continued use in chimeric multiepitope protein constructs for more sensitive and rapid IgA diagnostic tests [64].

## 5. Conclusions

While some prior studies have identified IgA epitopes of the spike protein from SARS-CoV-2 [23,29,31], this is the first study to provide a comprehensive list of the IgA epitome. Out of the total epitopes identified, most (32) exhibited cross-reactivity with proteins from DENV and other organisms, including coronaviruses. Specifically, only eight epitopes proved to be exclusive to SARS-CoV-2. Among these, six were located within the RBB, and notably, the Omicron variants (BA.1, BA.2, and BA.2.12.1) presented a distinct IgA epitope (SCo/18/huA) compared to the ancestral strain. Furthermore, the study outlines the development of a chimeric polypeptide with multiple epitopes tailored to target SARS-CoV-2 IgA specifically. Leveraging the high precision of our RBD-based pELISA, it emerges as an invaluable tool for investigating the IgA response in SARS-CoV-2 infections, seroepidemiological research, and assessing vaccine coverage. Finally, the carefully selected epitopes identified in this study can serve as a foundation for designing an intranasal multiepitope vaccine [65]. The peptides we describe are currently undergoing optimization and adaptation for high-throughput platforms [66]. This development shows promise for establishing reliable antibody detection methods to support informed decision making for clinicians, the public health community, policymakers, and industry stakeholders.

## 6. Patents

The antigenic peptides described in this study are protected under Brazilian and US provisional patent applications BR 10.2019.017792.6 and PCT/BR2020/050341, respectively, filed by FIOCRUZ. They may serve as a future source of funding.

## Figures and Tables

**Figure 1 vaccines-11-01749-f001:**
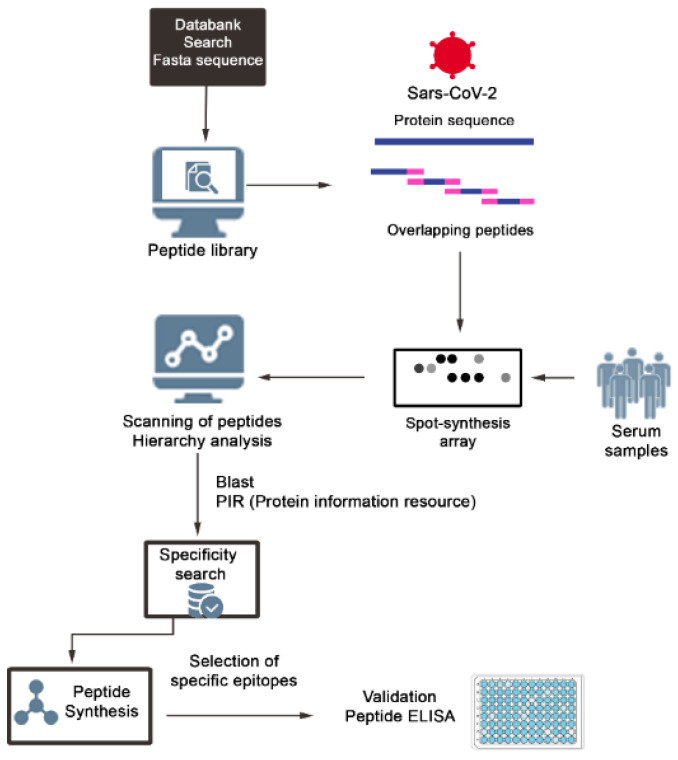
Schematic representation of the strategy used to identify and select SARS-CoV-2 IgA epitopes. Peptide libraries of 15 mer with overlapping of 10 residues were constituted using a software, and the positive peptides were revealed with patients’ sera and secondary labeled antibodies. The chemiluminescent positive spots were quantified using a scanner and those with the best performance were listed according with their hierarchic position using software. The possible cross-reactivity of the epitopes was checked in databanks, and those considered specifics were synthesized as single or multiepitopes peptides and validated by ELISA.

**Figure 2 vaccines-11-01749-f002:**
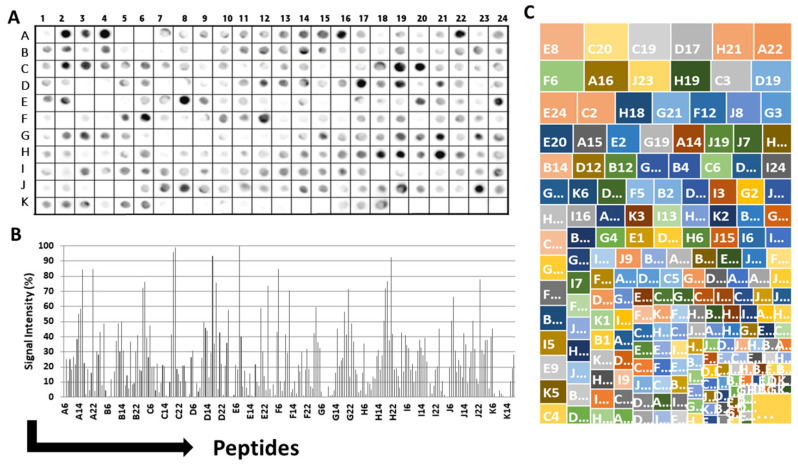
Interaction of human IgA with a cellulose-bound peptide library representing the COVID-19 Spike protein. An array of 144 overlapping peptides, each shifted by five residues, was immobilized on a cellulose membrane to express the COVID-19 spike protein. (**A**) The peptides were probed using a 1:250 dilution of a pool of human sera (*n* = 10), and alkaline phosphatase-labeled rabbit anti-human IgA and subsequent chemiluminescence detected human IgA. The membrane image displays the reactivity at each spot, and the designated positions on the membrane were used for the following measurements presented in the panel. (**B**) Relative signal intensity of bound human IgA at each position on the membrane. (**C**) An analysis of the hierarchical recognition of each epitope. The positive control established the reference point of 100%, whereas the negative control set the baseline at 0%. A comprehensive list of the individual peptides spanning the COVID-19 spike protein, which constitutes the peptide library, along with their corresponding positions on the membrane, can be found in Appendix A.

**Figure 3 vaccines-11-01749-f003:**
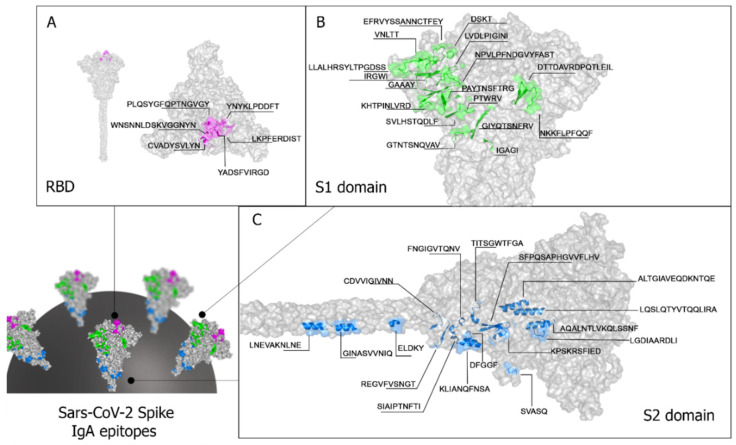
Spatial distribution of IgA reactive epitopes. The figure depicts the spatial positioning of IgA reactive epitopes within the receptor-binding domain (RBD) (**A**), the S1 domain (**B**), and the S2 domain (**C**) of the spike protein. No IgA epitope was identified within the protein’s transmembrane (TM) domains (aa1213–1237). The 3D conformation and structure of the protein were obtained from I-Tasser (https://zhanggroup.org/I-TASSER/, accessed on 28 July 2023).

**Figure 4 vaccines-11-01749-f004:**
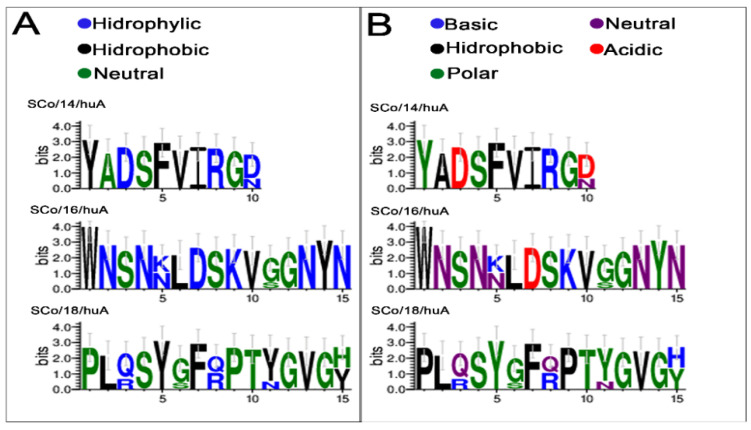
The sequence logo represents multiple sequence alignments. Visualization of the mutated IgA epitopes from SARS-CoV-2 spike RBD variants showing differences in charge (**A**) and amino acid composition (**B**). Each letter’s height is proportional to the frequency of the corresponding amino acid.

**Figure 5 vaccines-11-01749-f005:**
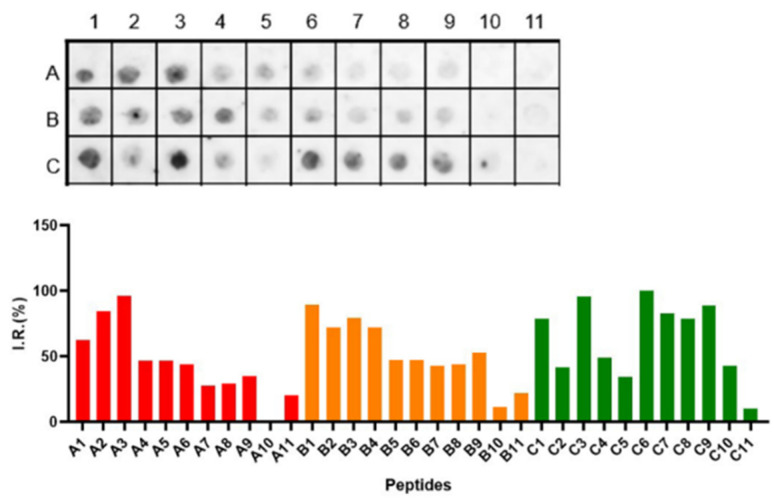
Performance of the mutated and non-mutated RBA IgA-epitopes as a target in a membrane synthesized peptide. The SPOT synthesis membrane immunoassay was prepared as described before to detect human IgA in sera collected during the first phase of the pandemic (A, red), after vaccination (B, orange; AstraZeneca), and when the mutated COVID-19 appeared (C, 2nd phase, green). Epitope SC/14huA (spot A1, B1 and C1; GGGYADSFVIRGDGG), epitope SC/16huA (spot A2, B2 and C2; WNANNLDSKVGGNYN), epitope SC/17huA (spot A5, B5 and C5; GGG LKPFER DISTGG). Epitope X (spot A7, B7 and C7; PLRSYSFRPPYGVGH), epitope Omicron BA.1 and Omicron BA2.12.1 (spot A8, B8 and C8; PLQSYGFRPTYGVGH) Omicron BA.4 (spot A9; PLQSYG FRPTYGV GH) and BA.5 (spot A10; PLQSYGFRPTYGVGH). The distribution of the synthesized peptides in the membranes is presented in Appendix A.

**Figure 6 vaccines-11-01749-f006:**
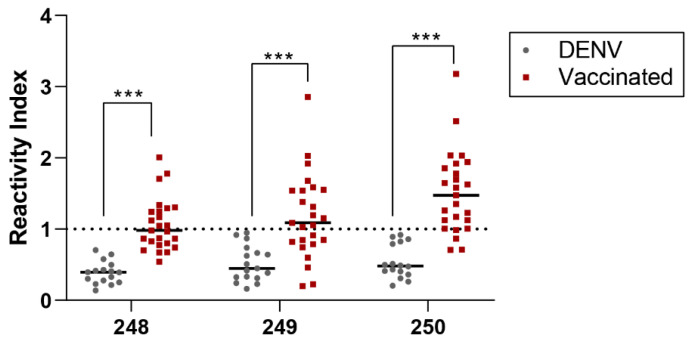
pELISA assays for IgA using Spike pep-248 (GSYADSFVIRDGGSG), pep-249 (GSNNSN NDSKVGSGS) and pep-250 (GSLKPFERDISTGSG) and sera from COVID-19 vaccinated (*n* = 25) and DENV-positive samples (*n* = 16; pre-pandemic). Data show the reactivity index of samples. Kruskal–Wallis’s test was applied to identify statistical differences, which was followed by Dunn’s multiple comparisons tests. Significant differences were considered with *p* < 0.05. *** *p* value < 0.0001.

**Table 1 vaccines-11-01749-t001:** Compilation of identified IgA epitopes in the SARS-CoV-2 Spike protein (P0DTC2) cross-reactivity. The SARS-CoV-2 spike protein encompasses epitopes numbered 1 to 40, which are recognized through SPOT synthesis. Subsequently, these epitopes were subjected to BLASTp analysis against non-redundant sequences from diverse databases. Amino acids aligning as hits are highlighted in red. Cross-reactivity was considered present when four or more amino acids aligned without gaps.

Code	Sequence	Aa Number	Domain	Cross-Reactivity	Organism
SC/01/huA	VNLTT	16–20	S1	Yes	DENV
SC/02/huA	PAYTNSFTRG	26–35	S1	Yes	DENV
SC/03/huA	SVLHSTQDLF	46–55	S1	Yes	DENV
SC/04/huA	NPVLPFNDGVYFAST	87–95	S1	Yes	DENV
SC/05/huA	IRGWI	101–105	S1	Yes	Several
SC/06/huA	DSKTQ	111–115	S1	Yes	Several
SC/07/huA	EFRVYSSANNCTFEY	156–170	S1	Yes	DENV
SC/08/huA	KHTPINLVRD	207–215	S1	Yes	DENV
SC/09/huA	LVDLPIGINI	226–235	S1	No	-
SC/10/huA	LLALHRSYLTPGDSS	241–256	S1	Yes	DENV
SC/11/huA	GAAAY	261–264	S1	No	-
SC/12/huA	GIYQTSNFRV	311–320	S1	Yes	DENV
SC/13/huA	CVADYSVLYN	360–369	S1/RBD	Yes	DENV
SC/14/huA	YADSFVIRGD	395–404	S1/RBD	No	-
SC/15/huA	YNYKLPDDFT	420–429	S1/RBD	Yes	DENV
SC/16/huA	WNSNNLDSKVGGNYN	436–449	S1/RBD *	No	-
SC/17/huA	LKPFERDIST	459–469	S1/RBD	No	-
SC/18/huA	PLQSYGFQPTNGVGY	490–504	S1/RBD	Yes	DENV
SC/19/huA	NKKFLPFQQF	555–564	S1	Yes	DENV
SC/20/huA	DTTDAVRDPQTLEIL	570–584	S1	Yes	DENV
SC/21/huA	GTNTSNQVAV	600–610	S1	No	-
SC/22/huA	PTWRV	630–634	S1	Yes	Several
SC/23/huA	IGAGI	665–669	S1	Yes	DENV
SC/24/huA	SVASQ	685–689	S2	Yes	DENV
SC/25/huA	SIAIPTNFTI	709–719	S2	Yes	DENV
SC/26/huA	ALTGIAVEQDKNTQE	765–779	S2	Yes	DENV
SC/27/huA	DFGGF	795–798	S2/Fusion *	Yes	DENV
SC/28/huA	KPSKRSFIED	810–819	S2	Yes	DENV
SC/29/huA	LGDIAARDLI	840–849	S2	Yes	DENV
SC/30/huA	TITSGWTFGA	880–889	S2	No	-
SC/31/huA	FNGIGVTQNV	905–914	S2/HR1 **	Yes	DENV
SC/32/huA	KLIANQFNSA	920–929	S2/HR1 **	Yes	DENV
SC/33/huA	AQALNTLVKQLSSNF	955–970	S2/HR1 **	Yes	DENV
Sc/34/huA	LQSLQTYVTQQLIRA	1000–1014	S2	Yes	DENV
SC/35/huA	SFPQSAPHGVVFLHV	1050–1064	S2	Yes	DENV
SC/36/huA	REGVFVSNGT	1090–1099	S2	Yes	DENV
SC/37/huA	CDVVIGIVNN	1125–1134	S2	Yes	DENV
SC/38/huA	ELDKY	1150–1154	S2	Yes	Several
SC/39/huA	GINASVVNIQ	1170–1179	S2/HR2 **	No	-
SC/40/huA	LNEVAKNLNE	1185–1194	S2/HR2 **	Yes	DENV

* RBD, receptor-binding domain; ** HR, highly conserved heptad-repeat region.

**Table 2 vaccines-11-01749-t002:** Multiple sequence alignment of identified IgA–RBD epitopes showing mutations (red amino acids) in Spike protein of SARS-CoV-2 variants of concern.

Variant of Concern	SC/13huA	SC/14huA	SC/15huA	SC/16huA	SC/17huA	SC/18huA
Wuhan	CVADYSVLYN	YADSFVIRGD	YNYKLPDDFT	WNSNNLDSKVGGNYN	LKPFERDIST	PLQSYGFQPTNGVGY
Alpha	CVADYSVLYN	YADSFVIRGD	YNYKLPDDFT	WNSNNLDSKVGGNYN	LKPFERDIST	PLQSYGFQPTYGVGY
Beta	CVADYSVLYN	YADSFVIRGD	YNYKLPDDFT	WNSNNLDSKVGGNYN	LKPFERDIST	PLQSYGFQPTYGVGY
Delta	CVADYSVLYN	YADSFVIRGD	YNYKLPDDFT	WNSNNLDSKVGGNYN	LKPFERDIST	PLQSYGFQPTNGVGY
Gamma	CVADYSVLYN	YADSFVIRGD	YNYKLPDDFT	WNSNNLDSKVGGNYN	LKPFERDIST	PLQSYGFQPTYGVGY
Omicron BA.1	CVADYSVLYN	YADSFVIRGD	YNYKLPDDFT	WNSNKLDSKVSGNYN	LKPFERDIST	PLRSYSFRPTYGVGH
Omicron BA.2	CVADYSVLYN	YADSFVIRGD	YNYKLPDDFT	WNSNKLDSKVSGNYN	LKPFERDIST	PLRSYSFRPTYGVGH
Omicron BA.2.12.1	CVADYSVLYN	YADSFVIRGN	YNYKLPDDFT	WNSNKLDSKVGGNYN	LKPFERDIST	PLRSYSFRPTYGVGH
Omicron BA.4	CVADYSVLYN	YADSFVIRGN	YNYKLPDDFT	WNSNKLDSKVGGNYN	LKPFERDIST	PLQSYGFRPTYGVGH
Omicron BA.5	CVADYSVLYN	YADSFVIRGN	YNYKLPDDFT	WNSNKLDSKVGGNYN	LKPFERDIST	PLQSYGFRPTYGVGH

## Data Availability

The data presented in this study are available upon request from the corresponding author.

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
