# Peer review of "Mapping IgA Epitope and Cross-Reactivity between Severe Acute Respiratory Syndrome-Associated Coronavirus 2 and DENV"

_vaccines, 2023, doi:10.3390/vaccines11121749_

Round 1
Reviewer 1 Report
Comments and Suggestions for Authors
Dear Editor,
I am sharing my review of the Manuscript ID vaccines-2687060 entitled: Mapping IgA Epitope and Cross‐Reactivity between Severe Acute 2 Respiratory Syndrome–Associated Coronavirus and DENV.
The article discusses the mapping of IgA epitopes of the S protein, along with its cross-reactivity, and the development of an ELISA-peptide assay. The research identified 40 IgA epitopes, with 23 in S1 and 17 in the S2 subunit, which showed cross-reactivity with DENV and other coronaviruses. The research also revealed that the S protein of variants from Wuhan to Omicron retains many conserved IgA epitopes, except for one epitope (#SCov/18). The identification of these specific epitopes as diagnostic biomarkers could facilitate monitoring mucosal immunity to COVID-19, potentially leading to more accurate diagnoses and alternative mucosal vaccines.
The manuscript should be accepted after minor revision.
The following issues should be addressed:
Introduction:
Line 55: …while IgG is the primary isotype in blood and most tissues.. which tissues, please give examples
Line 94: Please explain/describe the differences of COVID, DENV and CHYV
MM section:
Please add a figure describing the workflow.
Section 2.1 please add a Table with the patients description (gender, age, infection, clinical symptoms etc…).
Nice work!
Best
Author Response
1) Line 55: …while IgG is the primary isotype in blood and most tissues.. which tissues, please give examples
R: The sentence was corrected. “Among these, IgA, predominant in mucous
membranes, is the most abundantly produced Ig in humans (66 mg/kg/day), while IgG
is the primary isotype in blood and reaches most tissues by diffusion
2) Line 94: Please explain/describe the differences of COVID, DENV and CHYV. R: didn't understand the referee's question, but the sentence was modified.
3) MM section: Please add a figure describing the workflow. R: This new figure has been introduced. Please see in MM (Figure 1).
4) Section 2.1 please add a Table with the patients’ description (gender, age, infection, clinical symptoms etc…).
R: We believe that this table does not provide significant information for understanding the results of the work, despite this, in response to the referee's request, we introduced Table 1, with data from the study patients.
Reviewer 2 Report
Comments and Suggestions for Authors
The authors examined the reactivity and cross-reactivity of serum IgA in COVID-19 patients, using peptide arrays with overlapping SARS-CoV S protein sequence fragments of 14 residues. They mapped the identified epitopes onto the structure of the protein, analyzed their evolution in the viral strains and validated their use in ELISA.
Major Remarks
The authors state that this study is the first ever identification of the spike protein epitopes recognized by IgA. This seems to be an overstatement, as one of the cited papers also examined IgA binding to fragments of the S protein, and additionally there are several papers that are not cited even though they described IgA epitope analysis by various methods.
Examples:
DOI: 10.3390/v15010248
doi.org/10.3389/fviro.2023.988109
doi.org/10.1128/Spectrum.01416-21
doi.org/10.1038/s41467-021-21463-2
DOI: 10.3390/pathogens10040438
In order to assess the value of the submitted paper it is inevitable to include a thorough comparison and discussion of those previous findings.
Minor remarks
Under Materials&Methods the authors state they used 133 serum samples from symptomatic patients (COVID-19 ?), but Figure 1 presents results with a pool of 10 sera, then shows 33 sera in Figure 4, 30 sera in Figure S2, 25 sera in Figure 5. It is not clear how all the serum saples are related to each other, what and why was used for particular experiments.
Why do the authors think that IgA recognized epitopes would be markedly distinct from those recognized by other isotypes on a population level?
Author Response
1) The authors state that this study is the first ever identification of the spike protein epitopes recognized by IgA. This seems to be an overstatement, as one of the cited papers also examined IgA binding to fragments of the S protein, and additionally there are several papers that are not cited even though they described IgA epitope analysis by various methods. Examples: (10.3390/v15010248) (10.3389/fviro.2023.988109) (10.1128/Spectrum.01416-21) (10.1038/s41467-021-21463-2) (10.3390/pathogens1004 0438). In order to assess the value of the submitted paper it is inevitable to include a thorough comparison and discussion of those previous findings. R: We appreciate the information from the referee. At least 4 of the suggested Papers identified in various ways peptide fragments that may correspond to the IgA epitopes of the spike protein identified in our study. This information is being corrected in the text and references introduced. However, it is worth noting that none of the studies presents a complete list of Spike's IgA epitopes, unlike our work, and none attempted to evaluate cross-reactions with DENV and other Flavivirus. One of the studies assesses and discusses reactions with other coronaviruses (Camerini et al., 2021), another refers to a longitudinal study (Heidepriem et al., 2021), and another evaluates the Antibody affinity maturation of antibodies associated with clinical outcome (Tang et al., 2021) Furthermore, significant differences exist in the way/methodologies used to identify these epitopes. For example, the PepPer print technique (Acharjee et al., 2023) uses peptides printed in slides and single sera/plasma to identify epitopes. We have experience with this technique, and this approach is more useful for identifying individual variations in patient response than actually identifying the set of epitopes recognized by the immune system of a population. Compared to Spot-syntheses, this methodology is much more expensive and time-consuming, and the number of individuals to be analyzed needs to be greater to cover a broad spectrum.
2) Minor remarks: Under Materials & Methods the authors state they used 133 serum samples from symptomatic patients (COVID-19 ?), but Figure 1 presents results with a pool of 10 sera, then shows 33 sera in Figure 4, 30 sera in Figure S2, 25 sera in Figure 5. It is not clear how all the serum samples are related to each other, what and why was used for particular experiments. R: Thank you, there was an error in this sentence, there are not 133 but 33 serums. This was fixed in MM. Regarding the use of different numbers of serums in each experiment, we must inform you that the correlation of serums that the referee wants to make does not exist in the work, except that they are all from patients with COVID-19 collected in the first phase. In other words, several sera is used to identify epitopes [(Fig 2 and 5) (new)] which is followed by validation of the epitopes (Figure 4). The latter is even better to use different serums to validate the studies.
3) Why do the authors think that IgA recognized epitopes would be markedly distinct from those recognized by other isotypes on a population level? R: Because we have mapped all the 29 proteins of COVID-19 including all the isotypes IgA, IgM, and IgG (Patent pending)
Round 2
Reviewer 2 Report
Comments and Suggestions for Authors
Considering this previous question and answer:
3) Why do the authors think that IgA recognized epitopes would be markedly distinct from those recognized by other isotypes on a population level? R: Because we have mapped all the 29 proteins of COVID-19 including all the isotypes IgA, IgM, and IgG (Patent pending)
I think it would be useful for the readers and increase the scientific value of the paper if the authors briefly discussed why different antibody isotypes might have distinct epitopes or changed affinity for the same epitope or otherwise they referred to their own work (patent or paper) where they show how SARS-CoV-2 epitopes are different for IgA, IgM and IgG.
Author Response
As suggested, we broadly searched the literature on immunoglobin Swift. We introduced the information relevant to the discussion on our topic (marked in gray). Five more references were added, including our recently submitted work in which we evaluated the affinity of the IgG epitopes of the RBD of COVID-19,
We hope to have responded to the suggestion, and we would like to thank you for your commitment and evaluation, which undoubtedly improved the scope of the work.